# Should Artificial Intelligence Play a Durable Role in Biomedical Research and Practice?

**DOI:** 10.3390/ijms252413371

**Published:** 2024-12-13

**Authors:** Pierre Bongrand

**Affiliations:** Laboratory Adhesion and Inflammation (LAI), Inserm UMR 1067, Cnrs Umr 7333, Aix-Marseille Université UM 61, 13009 Marseille, France; pierre.bongrand@inserm.fr

**Keywords:** machine learning, deep learning, statistics, immunology, omic

## Abstract

During the last decade, artificial intelligence (AI) was applied to nearly all domains of human activity, including scientific research. It is thus warranted to ask whether AI thinking should be durably involved in biomedical research. This problem was addressed by examining three complementary questions (i) What are the major barriers currently met by biomedical investigators? It is suggested that during the last 2 decades there was a shift towards a growing need to elucidate complex systems, and that this was not sufficiently fulfilled by previously successful methods such as theoretical modeling or computer simulation (ii) What is the potential of AI to meet the aforementioned need? it is suggested that recent AI methods are well-suited to perform classification and prediction tasks on multivariate systems, and possibly help in data interpretation, provided their efficiency is properly validated. (iii) Recent representative results obtained with machine learning suggest that AI efficiency may be comparable to that displayed by human operators. It is concluded that AI should durably play an important role in biomedical practice. Also, as already suggested in other scientific domains such as physics, combining AI with conventional methods might generate further progress and new applications, involving heuristic and data interpretation.

## 1. Introduction

Artificial intelligence (AI) pervaded our everyday lives and scientific research [1,2], including domains as diverse as cell biology [3], medicine [4] or physics [5]; it is now included in the toolbox of many scientific investigators. Indeed, during the last decade, AI-related methods often denominated as machine learning (ML), including so-called neural networks or deep learning, has been tentatively used to support more and more numerous domains of scientific practice. In 2023, nearly 200,000 articles were retrieved on the Web of Science (https://www.webofscience.com/ accessed on 2 September 2024) by combining as keywords “artificial intelligence”, “machine learning”, and “deep learning”. In the widely used Pubmed biomedical database (https://pubmed.ncbi.nlm.nih.gov, accessed on 25 November 2024), the number of papers retrieved with the same keyword combination increased 27-fold between 2013 and 2023, respectively, amounting to 0.2% and 3.8% of all recorded papers. Thus, it seems now warranted to review the reported results and ask whether artificial intelligence should be durably involved in biomedical research and/or everyday practice. Also, since the widespread extension of artificial intelligence is relatively new, many readers with a biological or medical background might not be familiar with relevant terms and concepts, and it was found useful to recall basic definitions. Detecting possible causes of errors or misinterpretation is also of peculiar importance since, as discussed below, the wide availability and remarkable successes of AI facilitate its use by investigators with an incomplete knowledge of its limitations [6]. Finally, while this review is specifically devoted to the biomedical domain, it may be noticed that AI was also found to affect mathematical [7,8] and physical sciences [9]. Accordingly, this review was organized as follows:Selected examples of past research history were presented to support the view that reductionism, i.e., the analysis of simplified model systems, was partially replaced with more and more exhaustive analysis of complex systems, leading to the production of more and more extensive data sets that could not be adequately processed with previously successful tools such as mathematical modeling or computer simulation. This was an incentive to try and apply recently developed artificial intelligence tools.A brief description of current AI methods was then presented. Differences between AI and conventional statistics were discussed. Tools currently available to quantify the quality of results were then briefly described.Selected successes and failures of AI were then described to illustrate the points presented in the preceding two sections.

It is concluded that AI is bound to durably remain an essential tool of biomedical practice. Optimal benefit should be obtained by combining AI and human input. It should also be beneficial to investigate the inner workings of AI models, which should help us increase the reliability of conclusions, facilitate understanding, and suggest new experiments.

## 2. Identifying the Current State and Bottlenecks of Biomedical Practice

Due to the enormous diversity and heterogeneity of biomedical science, it was important to identify the tasks likely to benefit from AI. It was felt that a historical point of view might be useful in this respect; this might allow us to predict whether AI might generate a true revolution.

Arguably, major advances usually do not occur at random, but rather in a fairly ordered sequence, when scientific status is ripe to allow them. We shall try to assess this general concept by tentatively dividing recent research history into three sequential steps, which might clarify the present-day situation. A few selected examples will be described to allow an “informed guess” of future progress.

### 2.1. Setting the Stage: Basic Equations and Basic Description of Living Systems (1900–1950)

*Formulating the basic laws ruling molecular behavior*. During the first half of the 20th century, following a careful analysis of measured atomic spectra, quantum mechanics was elaborated by a number of investigators, resulting in a quantitative framework allowing successful calculation of atomic and molecular properties, and a theoretical interpretation (the so-called Copenhagen interpretation) that was questioned only recently [10,11]. It was then considered that the basic laws of nature were known. As stated in the preface of a celebrated quantum chemistry treatise [12], “chemical questions are problems in applied mathematics”. In addition to this fundamental domain, a quantitative framework was made available to account for basic phenomena such as intermolecular forces [13], chemical kinetics, or enzyme catalysis [14].

*General description of living organisms*. At the same time, biochemistry and microscopy were used to provide a general description of the molecular components and cell organization of living systems. Notable advances may be the discovery that nucleic acids could transmit genetic information [15], while the study of pepsin opened the way to the description of enzymes that were considered as major proteins playing a key role in the specificity of living systems [14].

*Conclusion*. At the middle of the 20th century, it had been accepted that biological systems followed the basic laws of physics and chemistry [16]. Further, thanks to a combination of experimental studies and remarkable theoretical interpretation, it was considered that these laws were known and liable to be applied to complex systems. Theoretical physicists were in some way considered as the heroes of modern scientific research. An influential book by Schrödinger [17] was an incentive for a number of physicists (including Crick [18]) to investigate biological systems with a highly quantitative point of view.

### 2.2. Towards a More and More Detailed Description of Living Systems (1950–1990)

During several decades, biology entered an era of rapid progress. As exemplified below, advances were driven by experimental advances associated with clever but fairly qualitative theoretical interpretations.

#### 2.2.1. The Molecular Biology Revolution: Discovering the Basic Mechanisms of Life

The elucidation of **DNA** structure generally ascribed to Watson and Crick is often considered as the foundation of so-called molecular biology [15]. The general mechanisms allowing genetic material to drive the development of complete organisms were progressively unraveled, as exemplified by the elucidation of the genetic code. The structure of **proteins** started being described with atomic resolution with X-ray diffraction [19], and functional properties such as enzymatic catalysis, conformational dynamics, and allostery were described. The importance of **cells** as fundamental units was accepted. Detailed structural studies were performed with light and electron microscopy and better fixation procedures. Basic signaling mechanisms were described, and the concept of **second messenger**, such as calcium [20] or cAMP, resulted in simple pictures of cell regulation [21].

#### 2.2.2. Attempts at Providing Detailed Accounts of Specific Processes, Leading to a Progressive Disclosure of the Complexity of Underlying Components and Mechanisms

Aforementioned general and schematic pictures acted as a starting point to more and more detailed descriptions of important phenomena. This may be illustrated by the following selected examples related to immunology, a domain that displayed state-of-the-art progress during the last decades of the 20th century and that remains deeply connected to cell biology and medicine [22]. Note that powerful methodological advances played a key role in this evolution. Thus, in addition to DNA manipulation techniques, monoclonal antibodies provided a way of labeling individual molecules in biological fluids or on the cell membranes [23] and radioimmunoassays allowed quantitative determination of the concentration of most molecular species with picomolar sensitivity [24].

*Achieving more and more precise characterization of cell populations: T-lymphocytes as a representative example*. Throughout all organisms, tissues are made of different cell populations, and a first step to understand organ function may be to identify all cell populations and determine their role. As a representative example, it has long been known that the entry of a foreign antigen into an organism is detected by the immune system that generates a fairly diverse response, including antibody production (the humoral response) and cell-mediated phenomena, hopefully resulting in the elimination of pathogens and infected cells. Investigations aimed at unraveling these phenomena resulted in the identification of several specialized immune cell populations, such as cytotoxic T-lymphocytes that were able to kill virus-infected cells, suppressor T-lymphocytes that were supposed to stop the immune response and prevent autoimmunity, or helper T-lymphocytes that were rapidly subdivided into Th1 cells enhancing cell-mediated response and Th2 cells enhancing antibody production. This example illustrates the degree of accuracy displayed in medical immunology textbooks that appeared in the 1980s [25]. Further, during the 1980s, the standard way of evidencing leukocyte subpopulations soon consisted of using flow cytometry to study cells labeled with a few monoclonal antibodies [26].

A systematic comparison of the binding specificities of antibodies raised by different teams led to the definition of so-called clusters of differentiation (CDs) grouping antibodies with a similar labeling pattern, suggesting that they bound to the same surface molecule. This led to a systematic compilation of leukocyte membrane antigens, as recapitulated in books published in the 1990s: about 200 leukocyte antigens were thus listed at the end of last century [27]. While the function of these antigens was first unknown, this could be discovered by studying the effect of monoclonal antibodies on cell function, or possible abnormalities of CD antigen expression in specific diseases. An early example is the identification of so-called lymphocyte function associated 1 (LFA-1, [28]), an adhesion molecule belonging to the integrin family, that was defective in Leukocyte adhesion deficiency (LAD) [29].

*Identification of cell adhesion receptors*. Cell adhesion is an ubiquitous phenomenon that influences most aspects of cell life. Since cells are surrounded by a hydrophobic membrane coated with negatively charged sugars, it was tempting to speculate that adhesion might be driven by the balance between van der Waals attraction and electrostatic repulsion following the well-known DLVO theory of colloid stability [30]. Further studies led to the hypothesis that protein-ligand interaction involving sugar-specific receptors called lectins and cell surface polysaccharides, might account for cell-cell adhesive selectivity without involving a considerable number of molecular species [31]. However, following the discovery of monoclonal antibodies in 1975, a growing number of cell surface molecules were progressively identified, and 108 adhesion molecules were listed in a compilation book that was published in the year 2000 [32]. Extensive structural studies led to the discovery of four main classes of adhesion receptors (integrins, cadherins, selectins, and cell adhesion molecules (CAMs)). Further experiments revealed that the efficiency and consequences of adhesion were determined by the nature of involved adhesion receptors and cell-activation state, thus warranting the detailed study of several adhesion models. Inflammation provides an important example.

*Inflammation* is a fairly universal process allowing higher organisms to fight against invading pathogens. This was dubbed “the backbone of pathology”, and its general mechanism was described more than a century ago [33]. A key step is the binding of flowing leukocytes to the endothelial walls of capillary vessels in inflamed regions, followed by transmigration into surrounding tissues. The essential picture was elucidated during the 1980s, as summarized by two important reports: (i) A detailed description of the initial interaction between leukocytes and endothelial cells was obtained with intravital microscopy [34] revealing sequential steps: flowing leukocytes first display a nearly hundredfold velocity decrease with a jerky motion called rolling, then stop after a displacement of several micrometers and finally transmigrate across the blood vessel walls (Figure 1). (ii) Adhesive events could be ascribed to specific adhesion receptors that had been recently characterized: capture and rolling were shown to be mediated by interactions between selectin molecules borne by the endothelial surface and specific receptors such as PSGL-1 on leukocytes. Firm adhesion resulted from a subsecond activation of leukocyte integrins such as LFA-1, resulting in strong interaction with their endothelial ligand ICAM-1. The rolling and adhesion phase were mimicked with an in vitro model [35]. A notable point is that this process involved the activation of rolling leukocytes by specific molecules (such as interleukin-8) on the surface of blood vessels. This gives us the opportunity to discuss the concepts of cell activation and signaling cascades that generated multiple studies.

*MHC restriction: a simple explanation for several puzzling observations*. It was for a long time difficult to understand why a T-lymphocyte specific for a given virus in an immunized host did not recognize the same virus on infected cells from another individual. The basic mechanisms were elucidated when it was found that antigens were only recognized in association with self-molecules [36]. Molecular details were investigated. The long-sought T cell receptors were characterized with tools generated by the molecular biology revolution [37]. It was unambiguously demonstrated that the target of T cells receptors was a complex formed between a major histocompatibility molecule and a short oligopeptide of about 10 residues resulting from the proteolytic cleavage of an antigen (Figure 2).

*T cell activation: a remarkable signaling cascade*. Cells usually perform their function only after they have been properly stimulated. As mentioned above, this triggering may often involve the generation of one or several so-called second messenger(s), such as calcium, induced by the engagement of a membrane receptor. *T-lymphocytes* arguably constitute one of the most convenient models of cell stimulation due to the possibility of monitoring individual cells under physiological conditions in a liquid environment. The signaling machinery was progressively dissected, leading to the description of an extensive sequence of reactions. The starting point is the formation of a multimolecular structure that was called a “signalosome”. About 10 molecular components were known in the mid 1990s [38].

*The cytokine network*. Investigations made on cytokines are well-representative of the progressive characterization of complex molecular systems that raised the need for sophisticated theoretical processing. We shall present here a brief summary, and readers are referred to [39] for more details. The concept of cytokines stems from early studies made on cell-mediated immune reactions that were found to involve biologically active molecules released by activated lymphocytes and accordingly denominated as lymphokines. Monoclonal antibodies and methods from molecular biology allowed progressive characterization of these substances at the molecular level. They were first thought to mediate leukocyte interactions and were thus denominated as interleukins. Thus, interleukin-1 was found to be released by *T-lymphocytes* and activate mononuclear phagocytes. However, it was rapidly reported that they acted on multiple molecular species. In addition, a given cytokine could be produced by several different cell species and influence different targets. In addition, cytokines displayed mutual interaction, leading to the concept of a cytokine network. More than 200 members were known at the end of the 20th century [39,40].

*In conclusion*, the systematic use of monoclonal antibodies and molecular biology methods yielded a more and more extensive description of the components of living systems, cell organization, and whole organism working. However, while theoretical insight played a key role in the molecular biology revolution, theoretical models were much simpler and limited than in physics: as written by the author of a remarkable account of this revolution ([15], p.xviii): “biology has been growing accessible to the general reader as it never was and as the modern physics never can be... molecular biology... is superbly easy to visualize”. This qualitative aspect of molecular biology was also pointed out in a Nature editorial: “Molecular biology seems well on the way to becoming a largely qualitative science. The notion that science hangs on measurement seems to have been diminished” [41]. Also, as emphasized by a discoverer of the DNA structure: “A theorist in biology has to receive much more guidance from the experimental evidence than is usually necessary in physics” [18]. This was in line with the remark that, “Charles Darwin laid down his great theory of evolution and the origin of species without making use of a single equation” [42].

### 2.3. Concomitant Attempts at Achieving a More and More Detailed Analysis of Small Molecular Systems, and at Building Sufficiently Exhaustive Data Sets to Provide a Holistic Description of Living Organisms: 1990–Today

The following two concomitant research strategies have been followed during the past 3 decades:*Carrying on reductionism. More and more quantitative dissection of mesoscale systems*. As exemplified in Section 2.1, studies made on simple models usually resulted in fairly qualitative pictures. This was clearly an essential starting point, but as formulated in cell biology textbooks in the beginning of the 20th century: “To progress from qualitative descriptions and intuitive reasoning to quantitative descriptions and mathematical deductions is one of the biggest challenges for contemporary cell biology” [43]. Accordingly, molecules and mesoscale (From the greek μεσοσ, meaning “intermediate”, models with a size intermediate between that of single molecules and whole cells or organisms. Mesoscale systems are being more and more thoroughly analyzed, as previously suggested [44]) molecular systems were revisited with increasing accuracy by leveraging newly developed physical methods such as cryoelectron microscopy, superresolution microscopy, atomic force microscopy, microfluidics, and so-called nanotechnologies allowing to prepare surfaces with nanometer-scale topographical control.*The omic era*. It was suggested that the reductionist approach that had met with much success would be insufficient to elucidate the function of living organisms since they involved complex networks of interacting molecules, resulting in the impossibility of defining the precise role of individual components [45]. A new strategy that might be called “systems biology” might thus be needed, with a requirement for both an exhaustive description of studied systems, which should be provided by so-called omic experimental studies, and a theoretical framework capable of processing the growing information flow. The need for new theoretical tools was clearly stated in a *Nature* editorial titled “Help, the data are coming” [46]. The idea that human intelligence might be unable to fathom this information was also clearly formulated in an editorial titled “Can biological phenomena be understood by humans?” [47].

The successes and failures of the above two strategies will now be illustrated with specific examples to identify current barriers and bottlenecks.

#### 2.3.1. Dissection of the Structure and Function of Small Systems with More and More Quantitative Accuracy

The basic hypothesis supporting reductionism is that the understanding of a complex organism may be derived from a detailed analysis of its components. We shall now assess the validity of this concept by examining the advances achieved by going on studying the problems described in Section 2.2.2.

##### Adhesive Interactions and Receptor Properties

As previously described, leukocytes subjected to a hydrodynamic flow roll on a selectin-coated surface without stopping. In contrast, while rapidly moving cells do not interact with integrin-coated surfaces, they form adhesive bonds resisting flow after a brief static contact. This finding emphasized the biological importance of kinetic parameters in addition to thermodynamic affinity [48]. Another point is that an exhaustive knowledge of the adhesion receptors borne by two cells encountering each other is not sufficient to allow us to predict adhesion formation and subsequent duration. As detailed in a recent review [49], predicting the outcome of the interaction between cells bearing adhesion receptors requires a quantitative knowledge of the dependence of the bond formation rate on intermolecular distance and the dependence of the bond dissociation rate on applied forces. Interestingly, several different experimental approaches were independently elaborated to monitor the behavior of individual molecular bonds subjected to exogenous forces [50]. The *biomembrane force probe* may be viewed as an improved atomic force microscope that gave rise to a method of analysing intermolecular bonds that was called dynamic force spectroscopy [51]. This allowed one to interpret an intriguing phenomenon reported in a study made on the rolling phenomenon [52]: Selectins were found to generate adhesive interaction under flow, not under static conditions. This led to the discovery of so-called “catch bonds” that display an increased lifetime in the presence of (moderate) disruptive forces, in contrast with the usual “slip bond”, the lifetime of which is decreased by pulling forces [53]. Interestingly, many theoretical attempts were done at interpreting catch bond behavior with different methods such as molecular dynamics [54,55] or theoretical modeling [56].

##### Early Steps of *T-lymphocyte* Activation: 1—Correlation Between TCR/Ligand Binding Properties and Stimulation

A remarkable property of the TCR is its capacity to discriminate between its cognate pMHC ligand and tens of thousands of molecular complexes differing by a few or even a single aminoacid on the surface of a cell. Here, we shall exemplify recent research strategies by describing experiments performed to test two non-exclusive hypotheses.

(i)Lymphocyte activation might be due to a TCR conformational change triggered by the binding of a specific pMHC, in line with the well-known concept of allostery. While this hypothesis was not supported by cristallographic studies [57], it was emphasized that X-ray cristallography only revealed equilibrium conformations, while molecular dynamics evidenced transient conformations that might play a role in subsequent molecular interactions [58,59].(ii)The TCR decision might be determined by the physical parameters of TCR-pMHC interaction. Indeed, starting in 1994 [60], much evidence suggested that the interaction lifetime was the main determinant of activation. However, a few reports supported the view that the binding affinity was more important [61]. A possible means of resolving this apparent discrepancy was provided by an experiment consisting of manipulating a pMHC-decorated surface towards and away from a single T cell with simultaneous monitoring of the interaction force (with a biomembrane force probe) and intracellular calcium concentration (with a fluorescent calcium probe) [62]. A quantitative analysis suggested that a calcium rise was triggered when binding lasted more than about 10 s during a time period of 60 s (Note that this finding is consistent with the earlier report that calcium rise and phosphorylation events might occur a few seconds after antigen encounter [63] if it is considered that T cells can integrate signals provided by several TCRs as demonstrated by recent experiments [64]). Comparison between several peptide species allowed the authors to conclude that the stimulating activity of a pMHC was associated with the capacity of forming a catch-bond with the TCR. This conclusion was confirmed by a later study made on more than 50 different peptides [65]. However, this may not be the whole story: As previously emphasized, the “true” activation parameter is obviously related to what the T-lymphocyte feels, not to the result of physical measurements [44]. There is thus a need to understand how a T cell might feel the existence of a catch bond. A simple explanation might be that activation is triggered when the TCR is subjected to a force of suitable intensity during a sufficient time, thus triggering a specific conformational change. However, we cannot rule out the possibility that a T cell might integrate both the force exerted on the TCR and the loading rate (i.e., the time-dependence of bond-applied force). This interpretation is in line with a further report showing that the dependence of the bond strength on its age may also be a good predictor of peptide-activation strength [66]. The authors of a recent review [67] commented on the “considerable mystique” generated by the catch bond concept and suggested that a study of energy landscapes might provide a possible way of interpreting experimental data. Further, molecular dynamic simulations supported the hypothesis that force-dependent TCR conformational changes might play a role in cell activation [55]. As recently reviewed [68], elaborating a mathematical model of T cell stimulation remains a current challenge. It may not be unreasonable to speculate that machine learning tools might help us build a classifier integrating all binding properties measured on TCR/pMHC couples.

##### Early Steps of T-lymphocyte Interaction: 2—Initial Mechanism of Signal Generation

It must be emphasized that most biological processes are determined by parameters much more complex than the binding properties of a single ligand-receptor couple. Indeed, the outcome of T cell activation by a ligand-bearing surface is dependent on multiple factors such as the T cell state, dynamics of interacting surfaces, local membrane curvature and rigidity, and nature of membrane molecules surrounding the TCR [69,70]. As a consequence, current reductionist experimental studies are based on the development of well-defined models of cell stimulation with exquisitely controlled conditions, leading to the development of “platforms” allowing accurate control of nanoscopic details. The starting event is the phosphorylation of tyrosine residues located on the cytoplasmic part of three dimers (γε, δε, ζζ) composing the CD3 complex, which is physically associated with a TCR (Figure 2). Phosphorylation events are mainly mediated by p56lck tyrosine kinase that may be free or bound to a so-called co-receptor (CD4 or CD8, depending on the T-lymphocyte type). The phosphorylation state of tyrosine residues is determined by a balance between the activity of kinases, such as lck and phosphatases such as CD45, a bulky membrane molecule of about 45 nm size, which is about threefold higher than the TCR size. This triggers a signaling cascade starting with the recruitment of another kinase (ZAP70). Several non-exclusive mechanisms were suggested to account for the consequences of TCR engagement [38,44], and we shall only consider two hypotheses:According to the **kinetic segregation hypothesis**, bulky phosphatases might be excluded from regions of close contact between membranes bearing the TCR and its ligand, the total length of which is on the order of 30 nm [71]. This attractive hypothesis was subjected to more and more direct tests, as exemplified by the following two examples: (i) Live cells from the Jurkat line were deposited on anti-CD3-coated surfaces and examined with a combination of interference reflection microscopy and single-molecule localization microscopy, which allowed the authors to monitor CD45 segregation, activation events revealed by ZAP70 recruitment and cell spreading that occurred within a few minutes [72]. (ii) *T-lymphocytes* were deposited on nanoporous surfaces: Pores of 200 nm diameter, not 400 nm, induced both a separation of TCRs and CD45 molecules and signaling in the absence of the TCR agonist [73]. Similar results were obtained by trapping TCRs in regions where CD45 was excluded [74]. The kinetic segregation hypothesis was recently supported by simple mathematical modeling [75].The TCR might act as a **mechanotransducer**. As recently reviewed [76], during the last 2 decades, it appeared more and more clearly that cell function may be driven by mechanical as well as biochemical signals. An early example is the capacity of cells migrating on a surface to be guided by purely physical interactions and move towards stiffer regions, a phenomenon dubbed durotaxis [77]. The raising interest on mechanical cues was an incentive to subject TCRs to forces generated with anti-CD3 microbeads manipulated with optical tweezers [78]: A force on the order of 50 pN was found to trigger a calcium rise. This result might be of biological relevance since *T-lymphocytes* were shown to generate pushing and pulling force upon contact with anti-CD3-coated microbeads [79]. Also, T cells deposited on nanopillars bearing anti-CD3 antibodies generated forces on the order of 100 pN [80]. In line with these findings, molecular dynamics calculations supported the hypothesis that forces in the piconewton range might trigger conformational changes [81]. Hopefully, the link between forces and signal generation should be more thoroughly elucidated by combinations of traction force microscopy and more and more exhaustive imaging techniques [82,83].

*In conclusion*, this short summary illustrates the methods currently used to perform state-of-the-art reductionist analysis of the function of single molecules or small complexes in real time and under controlled conditions. In many cases, progress was achieved with fairly simple interpretations of experiments suggested by some clever and insightful reasoning. While they provided invaluable information on some basic principles underlying simple cellular events, it is unlikely that they might rapidly lead to an exhaustive account of all cell or organ functions in the near future for two reasons: (i) The complete elucidation of the working of a given molecular complex is dependent on multiple and incompletely known environmental features. (ii) Understanding the function of whole organisms on this basis would require one to identify and analyze a huge number of small molecular systems. As will be now discussed, these limitations were a strong incentive to follow a different research strategy. However, two points deserve to be added: (i) The aforementioned examples did not reveal any obvious need for current AI tools. (ii) While the aforementioned studies made on the immune system are arguably representative of many current models of interest, it must be emphasized that current attempts at elucidating basic brain mechanisms with a combination of functional imaging and standard psychological methods raise specific and altogether different problems that will be only briefly mentioned below.

#### 2.3.2. The Omic Revolution and the Attempts at Understanding Whole Systems

In contrast with the “bottom up” or “reductionist” approach, there was an increasing opinion that the understanding of biological systems required a complete elucidation of the interactions between their constituents since important functions could not be ascribed to a single molecular species [45]. This led to the development of so-called *systems biology*. The development of computer power and experimental methods supported the opinion that it should be possible to gather all information required to understand the working of a living organism and quantitatively use this information. For the sake of clarity, we shall sequentially summarize (i) the production of more and more extensive data sets and (ii) the search for adequate interpretive models. Hopefully, this will help us identify the problems that are most likely to benefit from the recent advances of artificial intelligence.

##### Experimental Advances: Building More and More Extensive Data Sets

*Genome*. The starting point of the so-called omic revolution was the *genome project*. Indeed, it was felt that the genome contained nearly all information required to build an individual, and the progress of DNA sequencing methods made it a reasonable goal to sequence the whole human genome [70,84]. This was rapidly extended to the “1000 genome project” [85] that was supposed to help interpretations by allowing comparisons among different alleles. Also, the sequencing of the genomes of more and more numerous living species was expected to help us understand the mechanisms of evolution and the genotype-phenotype relationship.

However, attempts at interpreting the data generated by these projects revealed the need for more and more extensive data sets, as briefly explained below:

*Transcriptome*. Clearly, the differences between two cell species belonging to a same organism cannot be understood by considering the sole genome. The dynamics of protein synthesis is an essential determinant of the development and function of living cells, leading to an interest in studying the so-called “transcriptome”, with RNA analysis. As an example, an atlas of protein-encoding transcriptomes of 79 human tissues and 61 murine tissues was reported 2 decades ago [86]. This revealed the need for overcoming two limitations: (i) Due to the important heterogeneity of cells within an organism, it appeared important to perform single-cell RNA sequencing (scRNA-seq). Experimental methods have been efficiently elaborated [87], making scRNA-seq widely available to perform tasks such as a refined definition of cell populations [88]. (ii) In addition, another limitation of RNA sequencing was a consequence of the destruction of processed cells, which prevented any monitoring of the transcriptome dynamics at the single-cell level. Theoretical modeling was thus required to estimate parameters such as so-called RNA velocity [89]. A method overcoming this methodological problem (dubbed Live-seq [90]) was recently reported and used to monitor the “differentiation trajectory” of macrophages subjected to lipopolysaccharide stimulation.

*Proteome*. The function of a cell is dependent on its protein content, and this is not fully accounted by transcriptomic data. This was an incentive to determine the protein content of different tissues with mass spectrometry. Thus, a proteome atlas covering 12,000 genes and 31 normal human tissues was recently reported [91]. While this was concluded to yield useful information on metabolism or diseases, it was not sufficient to account for the organization of single cells, leading to the recent development of single-cell proteomics [92]. A last point is that cell function is heavily dependent on post-translational modifications of proteins. A prominent example is the so-called phospho-proteome, accounting for details of protein phosphorylation that is a major mechanism of cell regulation. Improved methods allowed phospho-proteome study with cellular and even subcellular resolution [93]. However, it rapidly appeared that a “static” atlas of cell protein content would not be sufficient to help understand its function. This was an incentive to study the dynamics of the cellular proteome following controlled stimulation. As recent examples, the proteome and phospho-proteome of *T-lymphocytes* were determined at times zero, 2 h, 6 h, 8 h, and 16 h after stimulation. A total of 8431 proteins and 13,755 phosphopeptides were assayed. Artificial intelligence was used to process experimental data [94]. In another study, a robotized platform was built to assay cytokine production triggered by T-lymphocyte stimulation: 24 TCR/pMHC stimulating combinations were studied, and in each case the production of 15 cytokines released by individual cells at different times after stimulation was assayed, leading to about 280,000 numerical values. Machine learning was used to process this information [95].

*Interactome*. It has been emphasized that the main function of a protein may be to bind to other molecules [96], making it an important issue to identify all binding specificities of the components of a living system, which constitutes the so-called interactome. This might provide a basis for selecting small molecular assemblies deserving to be studied in detail, as discussed in Section 2.3.1. Also, predicting drug-protein binding (also called docking) properties is an important challenge for **drug discovery**. As previously reviewed [49], an atlas of protein interactions should include a number of parameters (such as affinity, kinetic parameters, mechanical properties, and binding requirements) to allow meaningful interpretation of the biological significance of listed interactions. As an example, a recently published map of the human binary interactome involving about 17,500 human proteins (about 90% of the protein-coding genome) included about 53,000 interactions [97]. Thus, informative data sets were prepared by studying the evolution of actual interactions of several proteins of known interest. These experiments involved a combination of affinity purification and mass spectrometry [AP-MS]. As an early example, proteins interacting with ZAP70, LAT, and SLP76 were determined up to 300 s after *T-lymphocyte* stimulation, revealing 112 interactions, a majority of which had not been previously reported [98]. More recently, a similar study performed on 15 proteins revealed at least 366 interactions involving 277 proteins [99].

*Epigenome*. A complete understanding of organism function also requires a knowledge of epigenetic (i.e., not DNA-encoded) parameters. Indeed, regulatory elements encoded in our genome contain similar patterns of sequence-specific transcription factor binding sites [100], and transcription is dependent on a number of properties such as chromatin accessibility [101]. Thus, an atlas of chromatin accessibility across 13 murine tissues was reported, yielding a catalog of nearly 400,000 potential regulatory elements [102].

*Metabolome*. It has long been recognized that cell function is influenced by metabolic status, as exemplified by studies made on lymphocyte activity [103], with possible consequences in pathological situations [104]. See [105] for a recent review.

*In conclusion,* impressive methodological advances allowed the production of extensive data sets, often presented as atlases or resources, opening the way to a global understanding of the function of living organisms. This generated a need to apply and develop theoretical methods allowing one to interpret these data and use them to predict or even manipulate the evolution of organisms or cells. In the next section, we shall describe currently used tools together with provided results, as an introduction to the advent of artificial intelligence.

##### The Omic Revolution: Theoretical Challenge

In contrast with detailed studies of simple models that could be interpreted in a fairly intuitive way even if some calculations were needed, the processing of huge data sets generated a number of new problems that seemed liable to benefit from the input of artificial intelligence. We shall now describe these problems and rapidly sketch conventional methods that were used to address them, which will provide a convenient introduction to the expected input of artificial intelligence.

*Data display*. The output of omic studies is usually made of huge spreadsheets. A first step to the analysis of experimental data is to display results as concisely and exhaustively as possible. Indeed, as written in the introduction of a statistical manual, “*descriptive statistics allows us to picture statistical information in a lively and understandable way*” (as freely translated from [106]). A number of standard mathematical or statistical tools have been used for decades to display experimental data. Thus, a set of univariate measurements is usually summarized by calculating the mean value and standard deviation of numerical results. Studies made on the relationship between two quantitative variables are usually displayed as lines, bar charts, or scatter plots [107]. Three-dimensional projections are less frequently used and less easy to read [108]. However, these methods are not suited to display data sets including thousands of variables or more.

A more qualitative way of displaying poorly quantifiable results consists of drawing so-called graphs [109] or networks [110] consisting of sets of nodes, or vertices connected by lines called edges or links. This was used in many reports on “omic” data [94,111,112]. The problem is that it is not easy to achieve an intuitive grasp of figures including many tens or even hundreds of nodes and links.

*Dimensional reduction*. A common way of summarizing highly multivariate data is to identify key parameters that are thought to contain essential information and may then be represented with conventional plots. Principal component analysis (PCA) consists of sequentially defining mutually orthogonal linear combinations of variables accounting for maximum variation in a vector space. Other procedures such as independent component analysis [113] or more elaborate methods such as UMAP [114] were also used [115]. The problem is that the significance of these components is not easy to understand, and there is some arbitrariness in their definition and/or selection.

*Cluster analysis* is a standard way [116] of summarizing complex data by assuming that they represent the superimposition of simpler populations or phenomena. Thus, splitting a heterogeneous cell population into a set of presumably homogeneous cell species is a first step to functional or structural studies. Flow cytometry or transcriptomics may be used for this purpose [117]. Clustering may be performed by visual examination when univariate or bivariate samples are studied. More elaborate methods such as *hierarchical clustering* can be applied to separate points in a multidimensional space when a distance has been defined.

*Use of standard mathematical modeling*. Standard mathematical tools were often used to try and achieve a quantitative account of well-defined cell functions. This strategy is illustrated and hopefully clarified by the following three examples:(i)*Ordinary differential equations* (ODEs) were used to provide a description of the interaction between 15 biological species in a virus-infected organism [118] (see [119] for additional examples). A problem with this method is that it is difficult to determine accurately the kinetic parameters involved in the equation and the initial parameter values of a system to predict its evolution.(ii)*Boolean networks* provide a semi-quantitative way of approximating the effect of multiple interactions between biological species with simple functions (such as “activates” or “inhibits”). As an example, Boolean networks and ODEs were used to study the stability of epithelial to mesenchymal transition [120]. See [49] for other examples and [121] for recent attempts at simulating complex signaling networks with a Boolean model.(iii)A framework ascribed to *Waddington* met with durable popularity in representing cell differentiation as the displacement of a marble on a hypersurface (see [119] for a brief summary). The cell state is represented by the marble coordinates, and these may represent the RNA values yielded by single-cell RNA sequencing. As a recent example, constructing a dynamical landscape from quantitative expression data allowed one to predict the fate of stem cells subjected to a combination of signaling factors [122]. Other authors used single-cell RNA velocity data to quantify the Waddington landscape and analyze cell-differentiation pathways [123]. However, it must be admitted that these reports remain fairly exploratory, and they are probably considered as too difficult to read in the biomedical community.

*Developing specific tools for dealing with complexity: systems biology*. Concomitantly with the development of “omic” measurements, attempts were done at developing a “science of complexity”, as illustrated by a number of textbooks [124,125,126]. Theoreticians suggested explanations for the appearance of so-called “emerging properties” in large molecular assemblies. It was hypothesized that biological networks followed fairly universal laws, the elucidation of which should clarify the significance of the aforementioned graphs. Thus, it was suggested that simple subnetworks denominated as motives might fulfill well-defined functions, as may be exemplified by OR or AND operations that are a basis of computer function [125,127]. Experimentally reported networks could then be analyzed by looking for motives with unexpectedly high frequency or fairly isolated subnetworks that might be considered as “modules” [128,129].

*Simulations*. A powerful way of studying molecular systems consists of building quantitative models as realistically as possible and simulating their evolution with a computer. Molecular dynamics (MD) [130] is probably the most important example, and it is currently used in combination with X-ray cristallography to study the equilibrium structure and dynamics of proteins. A limitation of this approach is that it yields large amounts of data that may require advanced mathematical tools for analysis [131], and that only short phenomena (i.e., lasting less than a few microseconds) can be fully simulated with currently available tools. However, attempts were made at overcoming this limitation with improved procedures. Recently, a combination of docking and simulation methods was reported to allow protein simulation at atomic resolution for a duration of order of a second [132]. The equilibrium of Rho GTPAses between an active membrane-bound and an inactive cytosolic pool was simulated with a reaction-diffusion model, leading to the conclusion that this process allowed cells to perceive their own shape [133]. Also, a fully dynamical model of a minimal bacterial cell with only 493 genes was described in a promising report [134]. While clinical trials are usually needed to assess the efficiency of new anti-tumor drugs, a model simulating the effect of immunotherapy on the treatment of liver cancer was built with published multiomic parameters [135]. However, it was estimated that efficient simulation of systems as complex as the human brain would not be feasible before several decades [136].

*Conclusion*. Many attempts are currently done at using sophisticated mathematical and statistical techniques to visualize, summarize, and model the huge data sets provided by omic experiments. In view of the difficulty not only of developing these models but also of understanding their significance and making use of them, it is not surprising that many investigators addressed these problems with models from artificial intelligence. As will be shown below, many currently available AI tools are fairly easy to use and display impressive versatility and capacity to process very complex data sets. We shall now provide a brief introduction to currently popular approaches and describe selected examples to give a feeling of the results currently obtained.

## 3. Brief Introduction to Currently Available ML Tools

Artificial intelligence was considered for more than 50 years as a mimicking of human intelligence that computers might perform in the future. While intelligence is complex and fairly difficult to define [137], a key property of so-called intelligent systems is their capacity to learn and adapt to environmental changes. Accordingly, researchers have long tried to implant this capacity in computers with many different approaches [138]. For the sake of clarity, it was felt useful to provide a brief definition of currently used terms and procedures.

### 3.1. Basic Definitions

Mathematics may provide us with an accurate language: As explained in a general treatise [139], “One of the most basic activities of mathematics is to take a mathematical object and transform it into another one. A **function** is, roughly speaking, a mathematical transformation of this kind”. As an example, an image (i.e., a set of pixels), such as a handwritten digit, may be transformed into a number by a computer (“reading”) program. This program is an artificial intelligence tool that may thus be viewed as a function. This is currently called an artificial intelligence **model**. It may be entirely written by a programmer, which may require much time and fairly high skill. Alternatively, **machine learning** (ML) may consist of using a fairly sophisticated algorithm that allows **autonomous** building of the function that will transform handwritten character images into numbers. The aforementioned reading tool, or model, is built by processing a (fairly large) set of annotated handwritten digits that is defined as the **training data set**. Further, the set of labels (i.e., numbers represented by these digits) may be defined as the **target data set**. Building a model with a training data set and a target data set is called **supervised** learning. Using the language of Python, a computer language that may be considered as the lingua franca of artificial intelligence [140], the aforementioned sophisticated algorithm behaves as a **class** object. As clearly stated in a standard Python reference book [141]: “a class object is a factory for making instance objects”. The aforementioned digit-reading model may be viewed as an **instance** object. This is an implementation of the mathematical function object that transforms an image into a number.

Let’s take another example to go further: KNeighborsClassifier is a machine learning class provided by scikit-learn (an open access machine learning project: http://scikit-learn.org, acceded on 21 September 2024). This class will allow us to perform so-called nearest neighbor classification in a set of objects [142]. As a simple example, we may think of a training data set made of points labeled A or B in a defined region of space. Here, the method used to classify any new point P consists of determining the k nearest neighbors of P in the training data set and taking for label of P the label matching the majority of k nearest neighbors. Building a classifier with a given number of neighbors, say k = 7, may be achieved with the following two code lines, as written in Python, which seems quite self-explanatory:

knn = KNeighborsClassifier(n_neighbors = 7)

knn.fit(TrainingDataSet,TargetSet)

Thus, KNeighborsClassifier may be viewed as a superclass that will build class *knn* when we have chosen n_neighbors that may be called a **hyperparameter**. *knn* will then behave as a class that may be used to build a classification instance, or model, with the built-in function *knn.fit* that performs model **training**. The trained model may then be used to classify any point or array of points with the function *knn.predict*. It is important to understand that KNeighborsClassifier (as well as other scikit-learn classes) includes default values of a number of hyperparameters. Since these are usually very cleverly chosen, it is easy to forget their existence, and this is an important point of caution: Currently used artificial intelligence algorithms usually include a number of default parameters that may play a key role in end results, as well as the data set. Numerical examples of the importance of hyperparameters were provided in a recent report made on the use of basic ML classes to discriminate between lupus erythematosus and mixed connective tissue disease with 13 biological parameters [143].

### 3.2. Versatility of Machine Learning and Potential to Support Biomedical Science

The extraordinary versatility of machine learning tools certainly contributed to the impressive development and popularity of artificial intelligence. It was therefore deemed useful to provide a brief description of the potential of machine learning. We shall first describe functions that are related to statistics that has long played a key role in biomedical sciences. We shall then briefly discuss more recent applications, in particular those based on so-called generative learning.

#### 3.2.1. Machine Learning as an Extension of Statistical Science

Machine learning is often considered as an extension of statistics [144], and there is a wide overlap between both domains. Indeed, it has been clearly acknowledged by statisticians that the development of computers resulted in an extension of statistical science and the appearance of new fields such as data mining [142]. Thus, it was felt instructive to recall briefly what “conventional” statistics was used for before analyzing the added value of machine learning.

##### Scope of Conventional Statistics

“*Statistics deals with techniques for collecting, analyzing, and drawing conclusions from data*” [145]. Therefore, it is not surprising that statistical methods are used in most experimental studies. Thus, the standard way of summarizing measurements performed on a given variable in a population is to calculate the mean and standard deviation. Multivariate statistics is commonly used to represent statistical data expressed as numerical spreadsheets [106]. Statistical tests are commonly used to perform comparisons between a measured variable and expected value, for example, to achieve a medical diagnosis. Also, statistical tools have long been developed to perform operations such as classification, regression, or dimensional reduction.

*Classification*. Defining the state of a system may be defined as a classification problem. A general problem in medicine is to achieve a diagnosis and perform a **binary classification** between healthy and ill subjects on the basis of one of several variables. Common examples of **multiclass** separation are the identification of a cell as a member of a known population or the determination of the position of the cell in the cell cycle (with standard states of G0, G1, S, G2).

*Regression*. Regression is a statistical term that is commonly used in studies made on the relationship between two variables X and Y, as explained in a widely used statistical treatise ([145] p. 149): “*in mathematics, Y is called a function of X, but in statistics the term of regression of Y on X is generally used to describe the relationship*”.

*Dimensional reduction*. The appearance and development of computers generated a concomitant enrichment of statistical science and development of so-called “multivariate statistics” that was sometime used to describe spreadsheets including thousands of measurements and thousands of variables [106]. Accordingly, modern statistical tools include methods used for reducing the number of studied variables such as principal component analysis (PCA) or independent component analysis (ICA) that have long been used in the biomedical community [146,147].

*Clustering*. Looking for algorithms allowing one to find groups in data sets is an important field of research [116] at the frontier between statistics [106] and machine learning [148]. Determining clusters is an **unsupervised** task: clusters are determined in the absence of a target set, and there is some arbitrariness in this operation since there is no absolute way of determining the “true” cluster number and boundaries.

##### Specificity of Machine Learning

*Versatility*. Machine learning methods were built in part with well-known statistical tools such as logistic regression, hierarchical clustering, or decision analysis, and these were expanded into more efficient algorithms such as random forests or gradient boosting [142]. A next step was the development of entirely new methods such as neural networks that benefited from continual improvements, as illustrated by the appearance of more and more complex architectures exemplified by convolutional networks, recurrent networks, and transformers [149,150]. These displayed rapidly growing complexity, and a modern neural network model may include hundreds of thousands of fitted parameters or more, thus allowing one to produce more and more flexible and complex outputs.

*Autonomy*. It has long been felt that statistical tools were quite difficult to use and required a fairly high level of mathematical proficiency as well as access to specialized computer programs. This situation was somewhat alleviated by the development of R language [151], but this still required the acquisition of fairly specialized skills. In contrast, as exemplified above, a number of machine learning algorithms may be easily operated with a few lines of Python code. **Image analysis** provides an excellent example of the difference between ML and statistical strategies: Microscopy is a basic tool of biology and medicine, and numerous attempts have long been made at automatizing microscopy-based diagnosis (see [152] for more details and a specific example). Initial attempts involved a two-step procedure: First, a number of so-called texture parameters were defined and measured to provide a quantitative description of studied images with a few to more than 1000 numerical values [153,154], i.e., much less than the hundreds of thousands of pixel intensities. Second, these data sets were processed with fairly standard statistical tools. The problem with this approach is that the elaboration of texture parameters might rely on high-level mathematics. In contrast, fully autonomous machine learning models might be easily built with ready-made algorithms. Convolutional neural networks met with much success in this respect and somewhat outperformed the aforementioned two-step approach in recent studies [155]. In ML language, image characteristics are called features, and it is concluded that ML algorithms perform both feature selection and analysis, resulting in a fully autonomous model development.

*Algorithmic efficiency*. The success of ML is largely due to the progressive elaboration of highly efficient algorithms. As an example, while simple neural networks were elaborated 8 decades ago [156], it is only about 2 decades since they gained sufficient efficiency to replace humans in simple tasks such as handwritten digit reading thanks to innovative algorithms such as back propagation or convolutional architecture [149]. It may be emphasized that this rapid progress might be ascribed at least in part to the increasing size of the developer community, and possibly to the organization of competitions such as ImageNet that were organized to compare the efficiency of different algorithms [157]. A final point is that success was also due to the efficient use of computing advances, such as the programming of graphical cards to perform multiple calculations in parallel, and the increasing size of publicly available data sets. Indeed, the power of ML is clearly due to the association of huge data sets, high computer power, and the development of algorithms efficiently leveraging these advances [140].

*Regularization*. Another difference between statistical and ML tools may be related to a subtle difference between the general goal of both fields, as illustrated by a simple example: While the aim of statistical regression is essentially to describe a data set (in line with the denomination of “descriptive statistics”), the aim of machine learning is rather to build a predictive algorithm. For this purpose, instead of optimizing the fit with experimental results and a learned function, so-called regularization is used to achieve a balance between the aforementioned fit and the “simplicity” of learned parameters. Remarkably, optimizing a regularization parameter may improve the predictive value of the obtained function.

*Randomness*. There is a key point that must be kept in mind when a ML model is evaluated: a model trained on a given data set is dependent on the data set, as expected, but in addition, repeated training on a same data set may yield different results since the training process is not fully deterministic. Thus, neural networks are usually created with randomly determined starting values for parameters (often called weights). Further, parameter fitting is a complex process that proceeds autonomously without being controlled. Due to the variation generated by this randomness, but also by the choice of the aforementioned hyperparameters and by the complexity of ML models that cannot be checked by human control, as can be done for simple mathematical formulas, quality control is of particular importance in ML. This point will be discussed in Section 3.3.

#### 3.2.2. New Prospects Brought by Generative Learning

The continual development of machine learning resulted in the appearance of more and more seemingly creative capacities such as so-called generative learning. A prominent example is provided by so-called large language models (LLMs) such as ChatGPT [158]. While these models are based on algorithms such as deep networks that have been mentioned above, there is a major quantitative difference with the aforementioned tools: While a classifier may be seen as a mapping of a (nearly infinite) space of data sets into a fairly restricted set of classes, an LLM will map the (also nearly infinite) space of potential queries into the (nearly infinite) space of answers. Further, there is an increasing complexity in the respective processing of a plain spreadsheet, a scientific text supposed to use an accurate vocabulary, and common language. Accordingly, training an LLM requires a much more extensive data set than a standard classifier. In some sense, so-called generative capacity may be viewed as an *emerging* capacity of highly complex deep learning algorithms. We shall now briefly list a number of potential usages of generative AI in the biomedical domain. For the sake of clarity, examples will be ordered according to increasing generality. Specific examples will be given in Section 4.

*Answering questions that may be formulated with scientific accuracy*. Artificial intelligence is now commonly used to answer specific and well-limited questions that may be precisely formulated. Typical examples are the prediction of the spatial structure of a protein of known sequence, the outcome of interaction between two given molecules, e.g., for drug discovery, or the determination of a protein with a desired function.

*Data mining*. The enormous increase of available data, particularly through the world wide web, but also due to the steady increase of the rate of scientific publication, generated a need for automatic information-gathering. The increasing capacity of ML tools to process scientific texts brought a major contribution to data mining and bioinformatics.

*Interpreting results of a scientific experiment*. Interpretation is usually considered as an advanced capacity of human intelligence. Only recently, limited attempts were used at performing clever processing of input experimental data to derive a theoretical model such as a set of equations [159].

*Assisting investigators with general tasks*. Recent examples are reviewing submitted manuscripts, generating scientific papers from experimental data, and writing grant applications (and conversely reviewing these applications).

*In conclusion*, the scope of AI appears more and more limitless. In Section 4, successes and pitfalls of recent attempts will be presented. However, we shall first describe general principles that must be known to assess the validity of AI-generated conclusions.

### 3.3. How Can We Detect and Avoid the Specific Pitfalls of ML?

It is widely accepted that refutability is a hallmark of science, as was first proposed by Popper [160]. Unfortunately, this general criterion is less and less liable to be applied as a consequence of the growing complexity of scientific theories. Indeed, even a mathematical theorem may be undecidable, as shown by Gödel [139], or it may be so complex that no mathematician will be available to check its proof [161]. Also, ML models such as deep neural networks were often compared to black boxes [162], the inner mechanism of which is difficult or even impossible to check. Thus, an essential step of the assessment of a scientific theory consists of using a sufficient series of checks to obtain a reasonably reliable conclusion. The methods currently used to assess the validity of ML models will now be rapidly described.

#### 3.3.1. Training, Testing, and Validating

The general procedure for supervised building of a ML model with an available data set consists of splitting the data set into a **training** and a **test** set. The training set will then be used to determine optimal values for parameters. This training operation usually consists of progressively improving a random parameter set by applying modifications resulting in a decrease of the difference between predicted labels and target labels. This difference is quantified with a so-called **loss function**. This “training” or “parameter fitting” is iterated until the gain brought by each step is “sufficiently” low, or if a default upper limitation of the number of iterations is reached. The trained model may thus be used to “predict” the labels of the test data set. The difference between predicted and actual values is quantified with a suitable metric, yielding the value that will be used as a reporter of model quality.

The validity of the training procedure may be hampered by a number of pitfalls. (i) The final set of parameters may be dependent on the initial parameter set and the validation procedure. In other words, this may represent a local minimum of the loss function in the parameter space. (ii) Due to the high versatility of many ML models, it is often very easy to achieve a very small loss after the training stage, but this may be due to the inclusion of parameters specific for the data set. This situation is called **overfitting**, and it has been found that a better prediction of the training set properties may be achieved if training is stopped before reaching the minimum of the loss function. This procedure was denominated as **early stopping** [163].

Another more subtle cause of error is the use of overlapping training and testing data sets to select an optimal model (by comparing different algorithms and fitting hyperparameters) and to estimate the model efficiency. This may result in artefactually high accuracy of trained models as a consequence of so-called **data leakage** [164]. This may be avoided by splitting a data set into a **training** set, a **test** set, and a **validation** set. The training and test data sets will then be chosen to train a number of models with different hyperparameter values and select the most successful combination. A realistic efficiency parameter can then be obtained by measuring the prediction accuracy on the validation set.

The aforementioned data leakage is also a major cause of limitation of the **generalizability** of a ML model. An obvious goal of ML scientists is to deliver fully trained models with an efficiency sufficient to assist scientific or medical tasks. However, it is often found that the actual efficiency of an ML model is markedly lower than initially estimated when it is applied to a new data set. This may be due to an artefactual use of irrelevant parameters. A well-known pitfall of medical trials leading to the need for careful randomization can be explained on a simple example: Suppose you wish to train a ML model to diagnose a given disease with a data set including a number of parameters measured on older patients and younger healthy controls. The model may be unable to detect a young patient. Why this risk is trivial: It may be less easy to detect if the distribution of a number of parameters (say: age, sex, weight, cholesterol level) is different in patients and healthy controls.

The aforementioned examples clearly emphasize the crucial importance of data set quality that is more difficult to control as data sets are large and provided by different procedures. In a recent report, **model disgorgement** was defined as the elimination of improper data and also of the effects of these data on a given model, and methods were suggested for this purpose [165]. A specific danger that was recently reported is the generation of large errors in models trained with data sets, including information produced by artificial intelligence; this was called **model collapse** [166].

An important—and not unexpected—point is that the quality of the training data set is an essential contributor to the efficiency of the trained model. Indeed, the impressive power of ML certainly stems from the possibility of using huge data sets [167], the quality of which is accordingly difficult to check. Conversely, it was noticed that ML is more “data hungry” than conventional statistics, resulting in low performance after training with an insufficient data set [168]. This is more a problem as “data hungriness” is higher in more powerful, multiparameter models. Two means of overcoming this limitation can be used. First, *data augmentation* [169] consists of supplementing a training data set with calculated data. For example, training a model for image analysis may be improved by adding to the training data set additional images generated by rotation or translation of the initial images. Also, *transfer learning* [170] consists of subjecting a model to an initial training with available data loosely connected to the specific situations that are addressed and *fine tuning* this model with a second data set matching more precisely the studied situation.

A recently emphasized problem is the so-called “*catastrophic forgetting*”: In contrast with the human brain, ML models subjected to multistep training may lose previous skills following a new training step [171,172,173].

#### 3.3.2. Metrics Used to Measure Model Efficiency

For the sake of clarity, we shall focus on the evaluation of a classifier efficiency: The simplest efficiency parameter is clearly **prediction accuracy** (*pa*), i.e., the fraction of accurate predictions. However, this parameter may depend on the composition of tested samples. Indeed, if the diagnostic efficiency of a binary classifier is estimated on a population comprising 90% of healthy subjects, a ML model predicting that 100% of samples are healthy will be ascribed a 90% prediction accuracy, which may be considered as fairly correct. Thus, a better and widely used parameter denominated as **kappa index** (*ka*) [174] was defined as:*ka* = (*pa* − *rpa*)/(*mpa* − *rpa*)(1)
where *rpa* is the prediction accuracy yielded by a random classifier and *mpa* is the maximum value yielded by a perfect classifier. In medical practice, it is often considered that the agreement is respectively moderate, substantial, or nearly perfect when *ka* is respectively higher that 0.4, 0.6, or 0.8. It must be emphasized that in any case, the significance of reported quality parameters is dependent on the quality of data sets used to determine them. Indeed, the capacity of a classifier to discriminate between patients and healthy controls is expected to depend on disease severity in the patients’ group.

Another method of assessing a binary classifier that is widely used in clinical practice consists of building the receiver operator curve characteristic (**ROC**) curve [29] that is used to account for the common situation where the diagnosis based on a quantitative parameter is dependent on a threshold value that determines the balance between the sensitivity (i.e., fraction of truly positive patients that are classified as positive) and specificity (i.e., fraction of truly negative patients that are classified as negative). The choice of this balance is guided by the consequences of the diagnosis. The ROC curve is a plot of sensitivity versus (1—specificity), and **the area under the curve** (*auc*) is used to account for the diagnostic value of a given model. In a recent review based on 56 published reports, the diagnostic value was considered as good or very good when *auc* was higher than a value ranging between 0.75 and 0.95 [175]. It must be emphasized that *auc* is dependent on the particular data set used for evaluation. Note that while parameters *ka* and *auc* are widely used in medical reports due to their high practical interest, they are also used in non-medical biological reports [176].

A final point is that there is certainly a need for the development of new methods to efficiently assess all aspects of AI [177].

## 4. Selected Example of ML Use

In order to illustrate the aforementioned concepts, selected applications of ML will be described. First, we shall sequentially describe attempts at unraveling the function of molecular systems, cells, and whole organisms. General tasks such as data mining will then be briefly discussed. Finally, some attempts at obtaining some theoretical insight into ML function will be described.

### 4.1. Contribution of ML to Molecular Studies

Detailed analysis of the structure and function of the molecules constituting a model system is a key step of any reductionist study.

*Protein structure*. The development of X-ray cristallography made it possible to determine the structure of biological molecules with atomic accuracy [19]. Starting from the elucidation of the myoglobin structure, an enormous experimental effort led to the determination of the structure of a growing number of proteins. However, as emphasized in 2021 [178], the nearly 100,000 protein structures recorded in protein databanks represented only a small fraction of the billions of protein sequences that were available thanks to the progress of molecular biology. Accordingly, much theoretical effort was done to derive protein conformation from sequence. A highly efficient approach called molecular dynamics (MD) consisted of simulating the motion of atomic assemblies with basic laws of mechanics and empirical potentials accounting for interatomic forces. The efficiency of methods allowing to derive protein structure from sequence was regularly assessed with the so-called Critical Assessment of structure prediction (CASP) biennal competition [179,180] consisting of comparing recently elucidated experimental structures to computed ones. As an example, conventional methods achieved less than 33% success in 2020, thus demonstrating the need for marked progress. An ML-based model called AlphaFold displayed spectacular improvement by winning the competition with 56% success [180]. This model involved a deep neural network and performed a highly efficient processing of available information, leveraging on the accepted conclusion that proteins include basic folds that appeared during evolution and were then incorporated in a number of different proteins. Excellent results were also obtained with Rosetta, another deep-learning-based model [181]. ML then became the dominant method in later competitions and AlphaFold rapidly improved prediction accuracy above 90% [182], thus illustrating the high improvement rate of ML models as compared to conventional methods. An important application of these advances is the capacity to design proteins with new shapes and functions [183]. The capacity of deep learning methods to design molecular complexes including not only proteins but also nucleic acids, small molecules, and metals was recently reported [184].

*Protein interactions*. As mentioned above, predicting protein interactions is an important challenge. Algorithms based on molecular structure have long been elaborated, and they were often more efficient than knowledge-based methods [185]. An important difficulty is that protein–ligand interaction may involve transient conformations that are different from the “equilibrium” structure. ML models were used to improve docking predictions along two lines. (i) A first approach consisted of combining ML and molecular dynamic calculations. Indeed, these calculations may generate large amounts of data corresponding to atomic trajectories, and ML was used to help process this information. As a recent example, the interaction between NFκB and DNA segments was studied with MD and ML models (logistic regression and random forests were used to process data generated by 148 µs MD simulation). This allowed one to identify residues playing a key role in total interaction [186]. (ii) Recently, successes were obtained with knowledge-based methods. It was reported that unrefined models of σ2 and serotonin 2A receptors provided with Alphafold2 might help predict interactions with ligands that had been experimentally identified. The authors suggested that Alphafold2 models might sample transient conformations involved in binding [187]. Note that a major use of the prediction of biomolecular interactions is in the support of **drug discovery**: As a prominent example, the increase of antibiotic resistance generates a need for the discovery of new antibiotics. Recently, a deep learning approach was claimed to predict the antibiotic activity and cytotoxicity of more than 12,000,000 compounds, allowing the authors to test 283 compounds that exhibited an antibiotic activity against *Staphylococcus aureus* [188].

### 4.2. Image Analysis

The development of convolutional neural networks (CNNs) with a remarkable capacity to recognize patterns [149] and, more recently, transformers with a capacity to account for context information [150,189] led to a rapid application of deep learning to routine and research image processing [190]. Image analysis was briefly mentioned above as a representative example of the capacity of ML methods to perform both feature extraction and image classification. Here, we shall briefly describe recent applications to biological research and medical practice. For the sake of clarity, we shall consider three situations: image enhancement, medical image analysis, and unconventional strategies.

#### 4.2.1. Image Enhancement

Enhancement of microscope images have long been performed with conventional tools [191]. Recently, convolutional networks were used to enhance immunofluorescence images by reducing background noise [192]. Other authors used ML to enhance the robustness of traction force microscopy, which consists of deriving the intensity of forces exerted by adherent cells from substrate deformation [193].

#### 4.2.2. Medical Image Analysis

Visual examination of microscopic images has long been performed to obtain useful information of biological or medical interest. However, this often requires specific skills and high amounts of time as compared to many modern analysis tools. This is especially inconvenient in medical practice, particularly when a rapid diagnosis is needed or when the price of a test may reduce its availability. Accordingly, newly developed ML models have been very rapidly and extensively applied to routine tasks. Here are a few examples:

Analysis of immunofluorescence images have long been considered as a major tool for the diagnosis of autoimmune diseases such as systemic lupus erythematosus [152]. Recently, deep learning has been successfully applied to analyze these images. As an informative example, a data set of 90,109 images was used to compare the diagnostic efficiency achieved by several convolutional networks and experienced or non-experienced readers: the best kappa index was 0.84 between the best CNN and experienced readers, while it was only 0.54 between a less experienced and senior readers [155]. See [152,194] for additional references.

Bone marrow trephine biopsy, a sampling technique preserving bone architecture, is considered as a crucial tool for the diagnosis of multiple myeloma. Recently, a deep learning model was reported to achieve diagnosis with an *auc* of 0.98 [195].

Analysis of pathological slides is a powerful tool for cancer diagnosis. Recently, a tranformer-based architecture was used to build a whole-slide analysis model that was trained on about 1.3 × 10^9^ images derived from about 170,000 whole slides. This was found to reveal mutations in several cancer types such as colorectal cancer and lung adenocarcinoma with an *auc* of order of 0.7, outperforming other models and allowing to assist clinical decisions [196].

A real-time analysis of needle biopsies is important for focal prostate cancer therapy. Recently, an innovative platform combining stimulated Raman microscopy and a deep neural network was shown to rapidly perform automated determination of Gleason score, a reporter of prostate cancer grade, yielding a kappa index concordance of 0.944 with a pathologist [197].

ML proved highly efficient at analyzing radiographs and may outperform experienced practitioners due to the capacity to exploit huge training data sets: A model trained on 147,000 thoracic radiographs outperformed standard risk scores for the prediction of cardiovascular events [198]. A model trained on 135,409 radiographs allowed one to detect fractures with an *auc* close to 0.97 [199].

Medical resonance imaging (MRI) is widely used to study many brain properties, and ML was used to analyze MRI images. As recent examples, a deep convolutional network (DCNNBT) was reported to achieve 99.18% accuracy in classification of four brain tumor types [200]. A slightly lower accuracy 96.03% was reported in a similar task of analyzing four-class MRI brain images with a combination of innovative feature extraction and support vector machine (SVM) classification [201].

A review of the medical literature published in 2019 led to the conclusion that ML, essentially convolutional networks, matched the efficiency of medical experts for diagnosis based on the analysis of medical images [202].

Finally, ML-assisted image analysis was reported to yield unprecedented help, such as a support to the identification of primitive tumors by analyzing metastatic cell images [203].

#### 4.2.3. Non-Standard Use of Machine Learning

The success of ML in replacing human expertise was an incentive to improve the overall efficiency of image analysis by **combining** human and machine potential. Such a combination was reported to improve the analysis of a set of images altered by artificial noise addition [204].

The aforementioned examples relied on supervised training of models on image data sets that had been classified by human operators. Super-resolution microscopy provides images that may be difficult to interpret, and human expertise may be lacking to annotate images. It was recently suggested that so-called “weakly supervised” paradigms might accelerate the exploration of nanoscale structures [3].

### 4.3. ML-Assisted Classification and Clustering

Classification and clustering are key steps of biomedical studies. A prerequisite to the understanding of a cell function is a determination of its differentiation state. Predicting the evolution of a given cell requires a determination of its initial state. Diagnosis is a first step of a patient’s management. These three situations will now be considered sequentially.

#### 4.3.1. Cell Types

As clearly stated [205,206], there is no standard definition of cell types, although it is critical for reproducible investigation. While a thorough discussion of this point would not fall into the scope of the present review, it seemed warranted to present a brief description of the methods currently used to recognize cell types in a heterogeneous sample on the basis of measured properties such as transcriptome, proteome, surface antigens, or functions.

Starting from the widely recognized idea that single-cell whole transcriptome determination is a gold standard for cell identification, a robotic data-acquisition system combined with deep learning was used to show that label-free cell imaging could efficiently discriminate between several *T-lymphocyte* populations [207]. This exemplifies the potential of ML to assist the identification of cell types with methods compatible with prolonged observation.

Flow cytometry has long been used for routine identification of immune cell subpopulations by rapidly measuring the expression of specific antigens on thousands of cells. While identification was first performed by visual examination of 2-dimensional plots built with a few highly specific markers such as CD4 or CD8, methodological progress made it possible to assay cells labeled with tens of markers. As recently reviewed [208], numerous ML methods were then developed for supervised or unsupervised cell identification. As a rather extreme example, more than 200 markers were used to resolve T-lymphocyte populations in different organs and under different conditions. This was performed with an analysis tools including UMAP for dimensional reduction and gradient boosting for classification [209].

In an attempt at grouping different cell types in a single framework, 38 data sets relative to 12 tissues were combined. A total of 55 immune cell types were defined out of a total of 119,046 cells [210].

#### 4.3.2. Analyzing the Complexity of Cell Behavior

An obvious prerequisite for the prediction of the future behavior of a given cell is the determination of its starting state. Although the concept of cell state has not been formally defined [211], this is currently used in a fairly intuitive way. For example, a cytotoxic T cell killing a cognate target it has just encountered is considered to shift from an exploratory state to active killing. A cell state may thus be viewed as a point or a region in a Waddington landscape, provided this formalism is considered as informative. We shall now describe two simple examples illustrating the importance of classification to understand cell function.

As described above, the state transitions of immune cells are in part driven by cytokines in a complex way. CD4+ *T-lymphocytes* were used as a model system to study signal integration: They were stimulated by numerous combinations of cytokines, and they were concluded to differentiate into a continuum of fates. Hierarchical clustering of cytokine groups was a basis to perform a mathematical modeling of signal integration [212].

A robotic platform was combined with machine learning to model the cytokine production displayed by CD8-positive cells after stimulation with different antigens: This allowed the authors to delineate six classes of antigens triggering different cell responses. Classification was performed with a simple neural network, using the scikit-learn platform [95].

#### 4.3.3. Classification and Clustering as Essential Tools in Medicine

Diseases and symptoms have long been defined by a fairly intuitive analysis of data provided by patient monitoring. However, it was progressively acknowledged that some kind of quantitative scoring should improve the reliability of diagnosis and therapeutic choice by adequate organization of clinical trials and statistical analysis. Indeed, nearly 30,000 references were provided on the Pubmed data base for year 2023 with keyword “evidence-based medicine” (https://pubmed.ncbi.nlm.nih.gov, accessed on 7 October 2024). Accordingly, as an extension of statistics, ML may be expected to be more and more frequently used in medical practice. Here are some representative examples:

An example of the application of standard statistical methods is provided by the derivation and validation of classification criteria for systemic lupus erythematosus about a decade ago [213]: A total of 702 patient scenarios and 17 biological or clinical criteria were processed with statistical methods including logistic regression and decision trees, resulting in an algorithm yielding a kappa index of 0.82 as estimated on 690 cases.

Rheumatoid arthritis is an autoimmune disease involving joint destruction. The effectiveness of treatments varies among patients, and it was hypothesized that this might be due to an heterogeneity of inflammatory conditions. This hypothesis was addressed by performing RNA sequencing on 314,000 cells from 79 donors, leading to the definition of six subgroups. Data processing was essentially performed with UMAP for dimensional reduction and standard statistical classification [214]. It may be emphasized that an improved disease classification is a first step to the development of **personalized medicine** [215], which is aimed at finding an optimal therapeutic strategy for each patient.

Mixed connective tissue disease (MTCD) was recently identified as a condition with some similarity to systemic lupus erythematosis (SLE) but with a different prognosis and treatment. Recently, a retrospective study was made on 44 patients to assess the capacity of basic ML tools to discriminate between MTCD and SLE on the basis of 13 biological parameters. Eight widely used classifiers were tested. An *auc* of 0.83 was obtained [143].

It was recently reported [216] on the basis of 149 scenarios that ML outperformed primary care physicians for so-called conversational diagnosis.

Thus, ML might fairly rapidly replace human intervention in elaborating and calculating diagnostic scores.

#### 4.3.4. Prediction

A major use of theoretical modeling consists of predicting the fate of a well-defined system in response to controlled stimulation. As described above, AI is well-suited to perform this task. We shall now describe representative examples of recently reported attempts, with an emphasis on reliability. As was clearly apparent in the preceding sections, medical practice is arguably the domain where trials were more numerous and prospects more important.

Schizophrenia is a major psychiatric disease. Starting from a comprehensive online resource for the adult brain built from 1866 individuals, a deep learning model was reported to improve disease prediction about sixfold as compared to available polygenic risk scores [217].

Atrial fibrillation is a frequently undetected heart dysfunction that may be associated with stroke and death. While conventional screening methods require prolonged monitoring, a database including electrocardiographs from 180,000 patients was used to train a convolutional neural network allowing to detect atrial fibrillation by analyzing standard 10s electrocardiographs with an *auc* of 0.87 [218].

An ML method based on support vector machine (SVM) was reported to predict cognitive decline in Parkinson’s disease on the basis of cerebral scans with an *auc* of 0.73 [219].

An ML model (based on random forests) was developed on 95,935 health records to build a coronary risk index on the basis of selected features (including 88 diagnostic codes, 104 treatments, and 81 biological data) yielding an *auc* of 0.95 for disease prediction [220].

As recently reviewed, ML is an increasingly used tool in oncology for diagnosis, prognosis, and treatment design [221]. As recent examples, convolutional networks and other ML algorithms were used to classify pathology slides for melanoma prognosis, yielding an *auc* of 0.8. An attractive prospect that deserves to be validated is the use of machine learning for dynamic adaptation of the treatment of cancer patients since personalized treatment may be the best way to deal with drug resistance [222]. Along a similar line, a mathematical modeling framework was suggested to predict immunotherapy outcome and thereby guide therapeutic strategies [135].

In the domain of synthetic biology, it was recently reported that a combination of ML and transfer learning could reproduce known protocols used for reprogramming cells with gene perturbation (typically knockdown or overexpression of a single gene) with an *auc* of 0.91 [176]. Along the same line, deep learning was used to design synthetic enhancers to target cells towards a given type [223]. Other authors used ML to build a tool (denominated as cSTAR) building on the aforementioned Waddington metaphor and ML (using the Support vector machine to relate cell types to regions of a multidimensional space). The aim was to use omic data to identify pathways converting one state into another [224].

*Conclusion*. Impressive and ambitious ML-based methods were recently developed to predict the outcome of the manipulation of living systems.

#### 4.3.5. Neurosciences

While neurosciences are not fully representative of mainstream biology due to the specific features associated with brain function, it was deemed necessary to mention this domain since (i) artificial intelligence is indeed designed to mimic human intelligence, and (ii) the study of neurosciences displayed an impressive expansion during the last 2 decades, following the development of functional imaging. However, two points deserve to be emphasized: (i) There are essential differences between brain and computer function. As an example, human memory decoding and recall are dependent on emotions, as demonstrated in studies made on hippocampus physiology [225]. (ii) Despite the aforementioned differences, a mechanism similar to that of convergent evolution [226] might be responsible for analogies between brain and computer algorithms used to perform a well-defined function. Below are a few representative studies:

Due to the nearly infinite number of possible images and the limited number of neurons available to encode them, it is reasonable to assume that only some specific features are encoded. This hypothesis was investigated by combining the measurement of neuron firing in specialized regions of macaque monkeys and ML-based adaptive image generation guided by neuronal response. Attempts were made at interpreting evolved images by comparison with 100,000 images retrieved from the ImageNet public database. The authors concluded that this study might improve our understanding of the dictionary of features possibly encoded in the cortex [227].

A deep convolutional network was used to analyze human face recognition, and it was concluded that this was well-explained as the result of optimization for the task of face recognition, and it did not fundamentally differ from the recognition of objects such as cars [228].

The capacity of deep convolutional networks to perform face recognition was compared to human performance with a database of more than 700 video clips. It was concluded that neural networks successfully captured human processes for categorical attributes of faces, not for dynamic processes [229].

This short and altogether incomplete description exemplifies the potential of ML to help analyze brain function.

#### 4.3.6. General Tasks of Potential Use for All Biomedical Investigators

We shall now briefly mention some examples illustrating the capacity of ML to impact many aspects of scientific practice independently of the specificity of biomedical sciences.

ChatGPT, a well-known large language model (LLM), was shown to improve the productivity of mid-level professionals engaged in writing tasks [230], and it was recently reported in a medical journal that more than 80% of authors had made use of an LLM such as ChatGPT to retrieve information, communicate scientific results, or write grant applications [4]. The broad use of ChatGPT in scientific writing was confirmed in a statistical study [231]. This may greatly enhance productivity. Indeed, it was estimated that researchers spent 60 days out of a yearly total of 260 workdays writing grant applications [232]. However, this might generate some problems relative to scientific reliability and integrity. Indeed, LLMs such as ChatGPT were reported to be able to produce fake data sets [233,234], and they were also found to commit errors in scientific tasks [235,236]. This is more disturbing, as ChatGPT was shown to be able to produce an article within less than hour after being fed with numeric data [237].

Thus, while ML and AI might strongly improve scientific productivity, it is important to develop control procedures [238].

#### 4.3.7. Perspectives: There Is a Need to Develop Additional Controls and Further Explore New Lines

While the examples presented in the preceding sections illustrate the impressive potential of present-state ML and are an incentive to apply currently available tools to try and solve biomedical problems, it is certainly warranted to look for innovative ways of increasing the reliability of current methods and expanding the range of applications. We shall now described recent investigations relevant to this objective.

##### Increasing ML Power and Reliability

The architecture of deep learning networks is often built by “trials and errors” strategies, and there is no general rule for determining the optimal number and size of hidden layers. While this may be guided by the intuitive knowledge acquired by experienced developers, it would certainly be beneficial to establish objective guidelines. As a recent example, an analysis of the relationship between classification efficiency and network depth led to the suggestion that high depth should be beneficial to classification, not to regression tasks [239]. Other authors reported an analysis of a simple situation supporting the interest of using deep networks [240].

The power of deep learning models is often ascribed to their capacity to learn features from data. This learning process is poorly understood, and it would certainly be useful to unravel involved mechanisms. Attempts at exploring this issue have been recently reported [241]. Recently, attempts were also made at understanding predictive failures [242] and generalization errors [68]. More generally, it has often been emphasized that ML models behave as “black boxes”. Deciphering their function should be informative in helping us understand the phenomena they are found to describe successfully [162,243,244].

It may be hoped that these attempts might progressively provide us with a general understanding of principles determining the efficiency and reliability of neural networks.

It is well-recognized that the power of ML models is in large part due to the amount of information brought by the huge data sets used for training. This led many authors to integrate fairly heterogeneous data sets, with a well-recognized danger of reducing the quality of training data [245,246]. Accordingly, several attempts were made at improving this situation. Thus, a method of integrating brain scans obtained with magnetic resonance imaging was elaborated and checked experimentally [247]. Methods suggested for integrating single-cell data sets were recently reviewed [248]. In line with this problem, an obvious but often forgotten cause of generalization errors is the so-called data set shift, a difference between data sets used for training and later application [249].

An important prospect would be to try and combine ML and conventional modeling, as was recently suggested [250,251].

##### Developing New Applications of ML

The progress of ML, and particularly the development of so-called generative artificial intelligence and large language models, were an incentive to develop new applications, as exemplified above. Perspectives for scientific tasks were discussed in a recent review [1], and only a few selected examples will be briefly mentioned below.

Refinement of experimental data is a process of immediate use that benefited from ML. Chromatin accessibility was mentioned as an important determinant of gene expression. This was an incentive to refine and check experimental data, as recently reported [252]. Fluorescence spectroscopy data were also improved with deep learning methods [253].

Seemingly highly creative applications were also reported: Deep learning was thus claimed to help derive partial differential equations [159] and assist the interpretation of complex physical models [254]. It was recently suggested that an adapted training procedure might allow standard neural networks to mimic human generalization capacity [255].

It may be noticed that so-called “creativity” may be viewed as a quantitative extension of classification rather than a qualitatively different process. Indeed, designing a new protein to fulfill a given goal amounts to selecting the most appropriate sequence from the huge—but not actually infinite—space of possible sequences. The same reasoning applies to the production of a human-understandable sentence. Thus, creativity may be seen as an emerging property of more and more powerful classifiers.

## 5. Conclusions and Perspective

Three important conclusions may be drawn from the data presented in this work.

First, the main conclusion is that **artificial intelligence should be durably involved in biomedical practice**, but the reasons for this conclusion are not the same in all domains, and two complementary cases may be considered:(i)Examples described in Section 4 suggest that ML might rapidly **replace** human operators in performing well-defined **routine tasks** such as image analysis, resulting in increased rapidity and decreased cost. Hopefully, this might result in increasing medical care availability in both high-income and less-developed countries, as was suggested [256,257]. Also, as exemplified in Section 4, AI might facilitate the development of personalized medicine. Indeed, while a personalized therapeutic strategy may be performed without any involvement of AI on the basis of a few or a single criterion, such as the presence of a specific genetic alteration sensitive to a particular drug in cancer ([29], p.460), AI might be used to process more complex sets of features such as metabolomic or proteomic data, thus substantially enhancing the scope of personalized medicine, which might be at the same time cost-effective and beneficial for patients. AI-mediated prediction, as described in Section 4.3.4, might play an important role along this line. Importantly, the aforementioned data suggest that it is already warranted to include AI methods in routine tasks where human operators are still needed. Indeed, (a) this should increase the safety of diagnoses or decision, since discrepancies between AI and human conclusions should raise the attention of experts on possibly tricky situations; (b) a systematic use of AI and comparison with established procedures would accelerate the evaluation of AI’s potential, and particularly of generalizability, which was shown to be an important issue.(ii)AI may be expected to foster the **progress of biomedical research** as a consequence of four congruent changes, as discussed in Section 2 and Section 3: (a) While the reductionist approach followed during decades was highly successful, the complexity of living systems became more and more apparent since seemingly simple processes such as cell adhesion or activation, or diseases such as cancer, were found to involve many tens of molecular components behaving as complex networks rather than linear sequences. (b) During the last 2 decades, more and more extensive experimental studies were performed, and generated data were made widely available as atlases, resources, or databanks. It may now be estimated that available information is already or should soon become sufficient to give a sufficiently detailed description of studied cells or whole organisms to predict their evolution. However, these data are no longer amenable to intuitive and fairly qualitative interpretation, as was the case until the 1990s. (c) Despite early predictions [46], the theoretical models built with standard mathematical and physical tools were at the same time unable to unravel the complexity of biological systems and too complex to be fully accessible to the biological community. (d) ML models were progressively improved due to a sequence of key discoveries. Thus, while AI was first viewed as fairly unsuccessful, it progressively approached human expertise, as evidenced by a number of competitions that were widely advertised. In addition, ML developers made their work as highly accessible as data providers. Thus, the congruence between the availability of large data sets and the impressive capacity of AI to process complex data sets supports the expectation that AI should generate substantial progress in biomedical research.

A second conclusion is that the versatility and complexity of ML methods imposes a special need to perform repeated checks of the validity of models in use. Indeed, in contrast with a mathematical theorem that may be definitively considered as reliable once it has been demonstrated, the validity of an ML classifier or predictor may be compromised following undetected changes of seemingly irrelevant features of processed samples. The possible consequences of this so-called data set shift were mentioned in Section Increasing ML Power and Reliabilit, and the limitations of ML reliability were discussed at length in Section 3.1. It may be noticed that the need for repeated testing of ML models with up-to-date control data sets is somewhat analogous to the need for systematically including controls in all series of samples that are subjected to any biological assay, even if the measurement protocol has been fully validated.

A third conclusion is that the versatility and rapid evolution of AI tools are an incentive to try and develop new applications, as exemplified in Section Developing New Applications of ML. It is hoped that this work will be an incentive for some biologists and physicians to consider incorporating AI in elaborating their research projects or in performing everyday tasks, and that at least part of the material described in this work will be useful in this respect. Due to the impressive pace of advances, it may be too early to review the definitive input of AI to biomedicine and predict whether this will behave as an important tool in some specific domains, or whether AI will generate a *bona fide* revolution, as was claimed to be the case for so-called molecular biology [15,258].

## Figures and Tables

**Figure 1 ijms-25-13371-f001:**
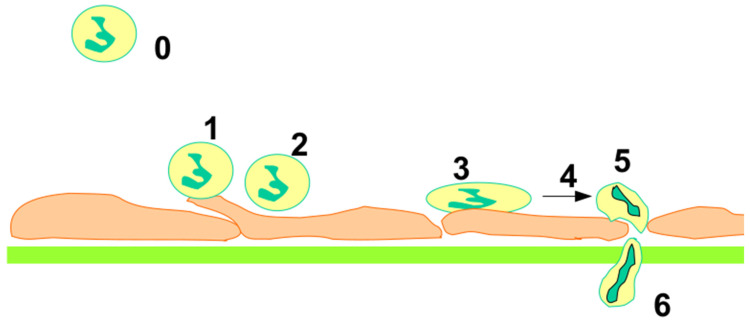
Early mechanisms of inflammation. A blood leukocyte flowing with a velocity of several hundreds of µm/s in a capillary blood vessel (0) is bound by selectin molecules expressed on the membrane of endothelial cells subjected to an inflammatory stimulus (1). This results in a nearly hundredfold velocity reduction, leading to a jerky motion called rolling (2). Cytokines bound to the membrane of activated endothelial cells then activate the surface integrins of rolling leukocytes, leading to a firm attachment to cell adhesion molecules expressed by endothelial cells (3). The leukocyte will then transmigrate towards surrounding tissues (4, 5, 6).

**Figure 2 ijms-25-13371-f002:**
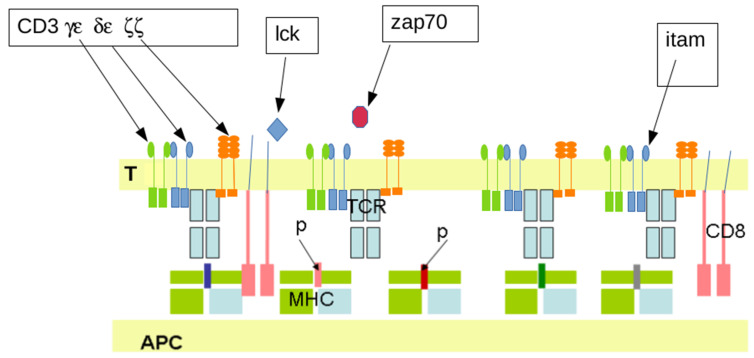
TCR-mediated T-lymphocyte activation. The figure shows a minimal view of molecules participating in the initial step of T-lymphocyte activation: An antigen presenting cell (APC) displays tens of thousands of major histocompatibility molecules (MHC) bearing various oligopeptides (p). The T cell receptor (TCR) is associated with the so-called CD3 complex typically made of three dimers (γε, δε, ζζ), the cytoplasmic part of which contains tyrosine-bearing motives (immunoreceptor tyrosine-based activation motifs called ITAMs, shown as colored ellipses). The association between a TCR and its cognate pMHC ligand enhances the binding of a co-receptor (such as CD8 on the figure) that is often associated with a tyrosine kinase (lck). TCR-pMHC interaction will thus trigger the phosphorylation of ITAMs by lck, and phosphorylated ITAMs act as docking sites for another kinase (zeta-associated protein of 70 kDa, ZAP70). Note this is a very simplified view. More details may be found elsewhere [38].

## Data Availability

This review paper does not involve any new data.

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
