# Peer review of "Should Artificial Intelligence Play a Durable Role in Biomedical Research and Practice?"

_ijms, 2024, doi:10.3390/ijms252413371_

Round 1
Reviewer 1 Report
Comments and Suggestions for Authors
Major :
1. In the Introduction and Current State Analysis of Biomedical Practice section,
The extensive discourse on molecular biology paradigms and immunological advances, while scientifically valuable, should be strategically condensed to better align with the manuscript's central thesis. Specifically:
- Consider streamlining the molecular biology content to emphasize only key developments directly relevant to the paper's main focus
- Synthesize the immunological advancements into concise, targeted examples that directly support your primary arguments
- Maintain essential historical context while prioritizing content that directly advances the manuscript's core objectives
2. In Conclusion Section, The discussion on future directions is limited and does not explore promising areas in depth, such as personalized medicine, drug discovery, or improvements in real-time diagnostics.
Minor :
"The manuscript requires standardization of the reference citation format. Please implement the following modifications:
- Citation Format Requirements:
All reference numbers must be enclosed in square brackets [#]
Citations should be positioned before the punctuation mark (e.g., [4]. or [1-3].)
For multiple consecutive references, use an en dash without spaces [1-3]
Multiple non-consecutive references should be separated by commas [1,4,7]
- Specific Locations Requiring Revision:
Line 25
Line 230
Line 345
[Additional line numbers where corrections are needed]
Reviewer 2 Report
Comments and Suggestions for Authors
Dear Authors
The paper titled “Benefit, prospects and limitations of artificial intelligence in biology and medicine” needs improvements.
1. The abstract has some drawbacks that need to be considered. It is quite vague and lacks specificity about the findings and methods used. The language is somewhat broad, using terms like "AI was briefly described" without clarifying what aspects of AI were actually discussed. The phrase "sketch of past research history" is unclear and doesn't explain what specific advancements or challenges in AI were highlighted.
2. Extensive English editing is required throughout the manuscript.
3. Section 1: Introduction: What is the novelty of this investigation????
4. The introduction also lacks focus; it briefly mentions different aspects of AI, machine learning, and deep learning but does not clearly explain their roles or impact. The mention of 200,000 articles retrieved from the web of science seems vague, without properly contextualizing this figure or its significance.
5. Section 3: The sections lack concrete, real-world applications to understand how these concepts apply to everyday tasks or problems. Furthermore, while the importance of certain parameters and techniques is mentioned, the risks of misusing them or how to mitigate those risks aren't sufficiently emphasized,
6. Section 4.3: This section should have what makes AI role in classification, and which type of recent applications it must justify its utilization in the present work. I suggest adding the following recent and relevant works as dcnnbt: a novel deep convolution neural network-based brain tumor classification model; brain tumor identification using data augmentation and transfer learning approach.
7. Overfitting and model tuning should be added as one section.
8. Limitations and the future scope should be added with more clarity.
Comments on the Quality of English LanguageExtensive English editing is required.
Reviewer 3 Report
Comments and Suggestions for Authors
ijms-3277346: Benefit, prospects and limitations of artificial intelligence in biology and medicine
By Pierre Bongrand
This research paper deals with the review on the current status of AI in the area of biology and medicine. It is concluded that machine learning should be durably involved in biomedical practice for data interpretation, in addition to classification and prediction.
*Completely rewrite the abstract, it is no good
*The introduction must be rewritten, in the current condition it does not identify the importance of the topic, the justification or any hypothesis. It must be radically improved
*It is necessary to indicate which methodology was used to carry out the bibliographic search. Keywords, years, database used.
*L10-11: The aim of the present review was to provide an easy-to-read basis allowing newcomers to address this task in the domains of biology and medicine. This is really the objective of the work, it is not congruent, you can publish another type of work that has that utility.
*L24: Artificial intelligence (AI) invaded many aspects of our everyday lives as well as scien-24 tific research [1] [2] , [3], [4] , [5]. It is not correct to agglomerate the references in this way. It is necessary to indicate the contribution of each of the references.
*L35 Typos: Recognized 35 pitfalls were also mentioned. this point is of peculiar importance since, as discussed below
*Although the article is structured in sections, some of them are too long and are not clearly differentiated, which makes the article confusingly structured and lacks coherence.
*The article is too general, introducing many areas of study without going into enough specific detail to support the statements.
*The title suggests that the article will focus on the benefits, prospects, and limitations of AI in biology and medicine. However lack of focus on artificial intelligence (AI): ,much of the initial content is devoted to topics that are not directly relevant to this focus, such as the historical evolution of molecular biology and physics.
*The conclusions of the article are unclear and unremarkable. The conclusions are very general and do not conclude with respect to the focus of the article's title.
*It is necessary to add a section of discussion of results, limitations, scopes and future work on the subject.
*The article is poorly structured and formatted. Some citations are poorly organized and unclear.
Dear author, the article needs significant revision to improve the quality, clarity, and depth of content before it can be considered for publication.
Reviewer 4 Report
Comments and Suggestions for Authors
Manuscript title: Benefit, prospects and limitations of artificial intelligence in biology and medicine
This paper provides a fundamental review on the application of artificial intelligent (AI) tools on the field of biology and medicine. However, the reviewer would like to reject the manuscript for publication for the following two main reasons:
1. This type of writing is more like writing a text book or some summary of lecture notes. As a result, this manuscript lacks significant new insights. For example, when discussing the pitfall of using machine learning methods on the subject of biology and medicine, the discussed topics, such as overfitting, data leakage, and etc. are not unique compared to the application of the ML methods in other fields of study.
2. The review has zero figure to illustrate any significant or seminal works in the context of ML. ML methods usually have their own architectural or logical designs for specific problems which can only be understood by logic flows. A review on ML without figures should not be published due to the lack of clear illustration. This lack of illustration also makes this review simply a pain summary of some previous works.
Round 2
Reviewer 1 Report
Comments and Suggestions for Authors
This manuscript presents a well-structured and meaningful contribution to the field.
Author Response
I thank the reviewer for his valuable comments and kind assessment of the revised version.
The paper was again carefully checked and, in addition to minor grammatical and stylistic corrections, the following three changes were done (as printed in red on the pdf version):
- the title of §2.2.2 was slightly extended to emphasize the purpose of suggesting that the increasing complexity of biological models spawned the current interest in omic and artificial intelligence.
- Two references were added to support the affirmation that dimensional reduction was used to address biomedical problems before the arrival of artificial intelligence (line 712)
- Since this article seems be planned to be inserted in a journal issue devoted to the medical applications of AI, it was deemed appropriate to insist on the interest of rapidly incorporating AI in routine medical tasks, even when reliability has not yet been fully validated, as explained in lines 1288-1294.
Reviewer 2 Report
Comments and Suggestions for Authors
All the comments have been addressed.
Author Response

(The authors gave the same response as above.)

Reviewer 3 Report
Comments and Suggestions for Authors
ijms-3277346: Benefit, prospects and limitations of artificial intelligence in biology and medicine By Pierre Bongrand
This research paper deals with the review on the current status of AI in the area of biology and medicine. It is concluded that machine learning should be durably involved in biomedical practice for data interpretation, in addition to classification and prediction.
Dear Author, I thank you for the improvements made to the manuscript.
I believe that the article needs a more rigorous and thorough review of its content, significance and contribution before being considered for publication.
Author Response
The paper was again carefully checked and, in addition to minor grammatical and stylistic corrections, the following three changes were done (as printed in red on the pdf version):
- the title of §2.2.2 was slightly extended to emphasize the purpose of suggesting that the increasing complexity of biological models spawned the current interest in omic and artificial intelligence.
- Two references were added to support the affirmation that dimensional reduction was used to address biomedical problems before the arrival of artificial intelligence (line 712)
- Since this article seems be planned to be inserted in a journal issue devoted to the medical applications of AI, it was deemed appropriate to insist on the interest of rapidly incorporating AI in routine medical tasks, even when reliability has not yet been fully validated, as explained in lines 1288-1294.
Reviewer 4 Report
Comments and Suggestions for Authors
The revised manuscript can be accepted for publication.
Author Response

(The authors gave the same response as above.)
